# Deceptive Problems Can Be Solved Without Explicit Diversity Maintenance

## Abstract

Navigating deceptive domains has often been a challenge in machine learning due to search algorithms getting stuck at sub-optimal local optima. Many algorithms have been proposed to navigate these domains by explicitly maintaining diversity or equivalently promoting exploration, such as Novelty Search or other so-called Quality Diversity algorithms. In this paper, we present an approach with promise to solve deceptive domains without explicit diversity maintenance by optimizing a potentially large set of defined objectives. These objectives can be extracted directly from the environment by sub-aggregating the raw performance of individuals in a variety of ways. We use lexicase selection to optimize for these objectives as it has been shown to implicitly maintain population diversity. We compare this technique with a varying number of objectives to a commonly used quality diversity algorithm, MAP-Elites, on a set of discrete optimization as well as reinforcement learning domains with varying degrees of deception. We find that decomposing objectives into many objectives and optimizing them outperforms MAP-Elites on the deceptive domains that we explore. Furthermore, we find that this technique results in competitive performance on the diversity-focused metrics of QD-Score and Coverage, without explicitly optimizing for these things. Our ablation study shows that this technique is robust to different subaggregation techniques. However, when it comes to non-deceptive, or "illumination" domains, quality diversity techniques generally outperform our objective-based framework with respect to exploration (but not exploitation), hinting at potential directions for future work.

## 1    Introduction

In the realm of machine learning, numerous problems exhibit inherent deceptiveness, *i.e.*, direct optimization towards a given objective within these domains often yields suboptimal solutions. To traverse these intricate problems, algorithms have been developed to place significant emphasis on explicit diversity maintenance, such as quality diversity algorithms (Pugh et al., 2016). Such strategies improve exploration capabilities, thereby avoiding the deceptive characteristics inherent in these problems. Nevertheless, a notable limitation of these explicit diversity maintenance techniques is the need to define and measure certain diversity metrics during the process of optimization.

In this work, we present an alternative approach to navigating deceptive domains without explicit diversity maintenance. We propose a method that leverages multi-objective optimization (MOO) to implicitly maintain stepping stones to global solutions. Instead of directly optimizing for a single objective, we decompose the objective into multiple sub-objectives and optimize for different combinations of these in parallel. This allows us to explore the search space more comprehensively, potentially uncovering high-quality solutions that explicit diversity techniques might skip. We utilize lexicase selection (Helmuth et al., 2015; Spector, 2012) to perform this optimization, due to its demonstrated implicit diversity maintenance properties as well as its superior performance in MOO. Lexicase selection selects individuals by filtering down a population based on a randomly shuffled ordering of the objectives.

Our experimental results demonstrate that our objective-based approach outperforms MAP-Elites on the deceptive domains we examine. By decomposing the objectives and optimizing them independently, we are able to achieve better solution quality and explore the search space more effectively.

Furthermore, our approach achieves competitive performance on diversity-focused metrics, such as QD-Score and Coverage, without explicitly optimizing for diversity. We also conduct an ablation study to assess the robustness of our subaggregation technique. The results show that different subaggregation methods yield similar performance, indicating the versatility of our approach.

Overall, our findings demonstrate the efficacy of implicit diversity maintenance using multi-objective optimization. By decomposing objectives and leveraging lexicase selection, we can effectively solve deceptive domains without the need for explicit diversity maintenance.

## 2 RELATED WORK

There exists a considerable literature of work surrounding finding solutions to deceptive problems. This work often is inspired by evolution in nature due to it being a fundamentally open-ended process. In this section, we outline the areas of research most relevant to this work, and provide references for a curious reader to learn more.

**Diversity-Driven Optimization**  Novelty Search (Lehman & Stanley, 2011a) was introduced as an algorithm to overcome highly deceptive fitness landscapes, where methods that follow objectives usually fail Deb et al. (1993). Lehman & Stanley (2011a) found that by "abandoning objectives" and simply optimizing for novelty in the solutions, deceptive problems can be solved more consistently. Since then, large research efforts have been lead into using Novelty search to solve previously unsolvable deceptive problems (Mouret, 2011; Gomes et al., 2015; Risi et al., 2010; Lehman & Stanley, 2010).

Novelty Search with Local Competition enhances Novelty Search's use by assessing individuals for both novelty and fitness (Lehman & Stanley, 2011b). This approach was among the first "Quality Diversity" (QD) algorithms, simultaneously optimizing for performance and diversity (Pugh et al., 2016). Another method optimizes aggregate performance and diversity with a multi-objective evolutionary algorithm (Mouret, 2011).

The MAP-Elites algorithm maintains an archive of diverse individuals with respect to specific qualitative "measures", with each cell housing individuals with the highest fitness found so far (Mouret & Clune, 2015). There have been many MAP-Elites variants that can exploit gradients (Fontaine & Nikolaidis, 2021; Tjanaka et al., 2022; Boige et al., 2023; Lim et al., 2023b), utilize advanced evolutionary algorithms (Fontaine & Nikolaidis, 2023), or apply policy-gradient-based techniques (Nilsson & Cully, 2021; Lim et al., 2023a).

QD algorithms typically solve deceptive problems, where there is a single optimal solution to find, and illuminative problems, where the goal is to span a series of qualitative "measures" or "descriptors" while maintaining high performance. A classic example of an illumination problem, that we consider in this work, is finding a set of reinforcement learning policies $\pi(s)$ that effectively reach a specific maze position $(x, y)$ while minimizing energy cost (Mouret & Clune, 2015; Cully et al., 2015).

**Multi-Objective Optimization**  The concept of multi-objectivation (Knowles et al., 2001) has been explored in traditional hill-climbing-style optimization methods to reduce the likelihood of becoming trapped in local optima. Our work further extends this concept with population-based optimization methods for implicit diversity maintenance, where the objectives are constructed according to different heuristics that may imply certain diversity characteristics. There has been a recent interest in combining Multi-Objective Optimization with Quality Diversity (Janmohamed et al., 2023; Pierrot et al., 2022). However, we consider large sets of objectives, which would cause dominance-based techniques like those in the literature to be ineffective due to the curse of dimensionality.

Another important aspect of this paper is lexicase selection, a popular parent selection algorithm developed for use in evolutionary computation systems (Helmuth et al., 2015; Spector, 2012). It has been shown to be effective at solving these problems when there are many objectives, as the domination-based MOOs suffer from the curse of dimensionality) La Cava et al. (2019). There also has been recent interest in comparing lexicase selection and quality diversity techniques (Boldi & Spector, 2023; Jundt & Helmuth, 2019). Lexicase selection continually filters down a population

---

**Algorithm 1:** Lexicase Selection for Implicit Diversity Maintenance

---

**Data:**

- `subaggregate()` - function that sub-aggregates raw scores

**Result:**

- A population of individuals

`candidates` ← randomly initialize $p$ solutions
**for** *iter* $1 \rightarrow N$ **do**
    `scores` ← subaggregate(Evaluate(`candidates`, `env`))
    `objectives` ← list of all objectives based on `subaggregate`($\cdot$)
    **for** *ind* $1 \rightarrow p$ **do**
        `shuffled_objs` ← Shuffle(`objectives`)
        **for** *obj in shuffled_objs* **do**
            `candidates` ← subset of `candidates` with exactly best performance on `obj`
            **if** *candidates contains only one single* `candidate` **then**
                **break**
            **end**
        **end**
        `candidate` ← a randomly selected individual in `candidates`
        Append `candidate` to new population
    **end**
    `candidates` ← mutate new population
**end**
**return** *candidates*

---

based on a random shuffle objectives by keeping individuals that are elite on the given test cases, in order. A more in depth description of lexicase selection can be found in Algorithm 1.

## 3 OPTIMIZATION WITH IMPLICIT DIVERSITY

Diversity-driven algorithms have been praised for their ability to explore complex search spaces by actively preserving phenotypic differences in a population of solutions, which is often achieved by employing explicit diversity maintenance, *e.g.*, MAP-Elites (Mouret & Clune, 2015). However, such explicit diversity maintenance has its limitations. It often needs prior knowledge to choose relevant diversity metrics. More importantly, in more complex or deceptive search spaces where the relationship between phenotypic traits and fitness is not straightforward, these explicit measures can inadvertently steer the search away from optimal or even satisfactory solutions. Take a deceptive maze that has no exterior walls. Novelty search and MAP-Elites might devote too much time exploring the space outside the maze as it is (rightfully) novel, whereas the implicit diversity preservation would only preserve diversity to the extent it helps reach the goal.

This work introduces implicit diversity maintenance as a promising alternative in such scenarios. Instead of overtly measuring and preserving diversity, our method inherently promotes diversity through its structure and operation. In general, we propose an objective subaggregation approach that turns the search problem into a multi-objective optimization (MOO) problem, and utilizes lexicase selection, which has been shown to produce diverse solutions and maintain such diversity, to solve it. This leads to a more adaptive and efficient exploration of the search space, potentially uncovering solutions that explicit methods might miss, especially when the terrain of the search landscape is intricate or misleading. An overview of the proposed method is described in Alg. 1.

### 3.1 OBJECTIVE SUBAGGREGATION

In the realm of general search problems, the objectives are often framed as aggregated values over a series of steps or stages. For instance, in reinforcement learning (RL), the goal typically boils down to maximizing the cumulative reward over time, *i.e.*, the sum of step rewards over the course of an episode, until termination. Similarly, in robotics problems such as navigation or control, the

overarching goal is often to reach a desired endpoint of a trajectory, determined by an accumulation of individual position changes over time.

Delving deeper into this notion of aggregation, we introduce the concept of "objective subaggregation". Instead of aggregating over all objective values, we focus on subsets of these values, where the subsets could be sampled based on certain heuristics or predefined rules. Given the set of steps or stages $T$ and $n$ heuristics, the $i^{th}$ heuristic $h_i$ samples a subset $S_i$ where $S_i = h_i(T)$. The sub-aggregated objectives $R$ can be expressed as:

$$R = \{R_i\} = \left\{ \sum_{t \in S_i} r_t \right\}, i = 1, 2, \cdots, n \tag{1}$$

where $r_t$ is objective value at step/stage $t$.

This objective subaggregation approach converts a single-objective optimization problem into a multi-objective or many-objective one, where the objectives can be used to promote diversity depending on the heuristics for sampling. This allows for more granular and targeted optimization that can potentially enhance the efficiency and effectiveness of the search process.

### 3.2 Multi-Objective Optimization with Implicit Diversity Maintenance

While the proposed objective subaggregation method can be potentially used to improve the diversity of solutions, another essential problem to solve is how to obtain and maintain such diversity during the optimization process. In this work, we utilize lexicase selection (Spector, 2012; Helmuth et al., 2015), which is a recently developed parent selection method in evolutionary computation. The idea of lexicase selection is that instead of compiling performance metrics over the training dataset, we can leverage individual performance to sift through a population via a sequence of randomly shuffled training cases. This strategy effectively tailors the selection pressure towards specific subsets of training cases that yield worse performance. Recent work also shows that by replacing individual performance with different objectives, lexicase selection can be used for multi-objective optimization, and outperforms state-of-the-art optimization algorithms (La Cava et al., 2019). A more in depth explanation of Lexicase Selection can be found in appendix **??**.

There are two main reasons for our specific choice of lexicase selection. Firstly, prior work (Helmuth et al., 2016) has shown that lexicase selection can maintain high diversity and re-diversify less-diverse populations. For complex search problems, the initial solutions (usually sampled from a Gaussian distribution) may not be diverse enough to drive the search, and may even lose diversity at certain points depending on the topology of the search space. Lexicase selection could potentially overcome these problems. Secondly, with different subaggregation strategies, the number of objectives for optimization after subaggregation may be large. Lexicase selection has shown to be the state-of-the-art method on many-objective optimization problems (La Cava et al., 2019) compared to other methods such as NSGA-II (Deb et al., 2002) and HypE (Bader & Zitzler, 2011) .

### 3.3 General Subaggregation Strategies

One important component in the proposed method is how the heuristics for subaggregation are chosen. We hereby introduce two general strategies that are used in this work and can be potentially extended to many other tasks in the RL, robotics, and beyond.

**Space-based Subaggregation**  Suppose there is some location information in each state $t$ that can be easily retrieved, *e.g.*, the position of the agent or end-point of a robot arm, we denote it as $(x_t, y_t)$ in a 2D space for simplicity. Each heuristic $h_i$ thus defines a sub-space $(X, Y)$, and construct a sample $S_i = \{t \in T \mid x_t \in X \text{ and } y_t \in Y\}$. An example would be to split the 2D space into $n \times n$ blocks, and each block is a heuristic to construct the sample. This is often a cost-free approach as long as such location information is inherently contained in the state variables.

**Time-based Subaggregation**  Given the trajectory of states $T$, we can define a subset directly based on its position. Here, each heuristic $h_i$ defines a sub-space $T'$, and samples from $S_i = \{t \in T'\}$. An example would be to split the whole trajectory into $n$ pieces where each piece is the sample for subaggregation. This is an even more general strategy that can be applied to almost any optimization and search problem.

## 4 DOMAINS

In order to test the problem solving performance of our implicit diversity maintenance technique, we consider two domains that are deceptive as search problems in general but vary in a few key aspects. We also present two variations of the latter domain to demonstrate robustness to task.

### 4.1 KNIGHT'S TOUR

A knight's tour is a path through a chess board taken by a knight that satisfies the following conditions: a) the Knight doesn't leave the board and b) Every tile is visited *exactly* once.

The Knight's Tour domain is inherently deceptive. There exist many local optima that are easy to be trapped in. For example, consider a knight's tour that visits all but 1 tile on the board. This knight's tour is very nearly optimal. However, reaching this solution might not be very helpful in finding the tour that visits all tiles on the board, as that solution would likely be largely different. This is a good domain to test the problem solving ability of quality diversity algorithms, as well as lexicase selection. The maintenance of stepping stones is also very important to this domain, so the use of quality diversity algorithms here fits their usual use case in deceptive problems.

**Objective and Subaggregation**    The objective in the knight's tour domain is to have the longest tour without breaking any rules. So, we define the fitness to be the size of the set of visited positions before the first rule break. The size of this set will range between 1 (worst) and 64 (best). This problem naturally lends itself to a space-based subaggregation where each heuristic $h_i$ references a quadtree decomposition of the positions of the board. For example, with 4 objectives, we split the $8 \times 8$ board into 4 smaller $4 \times 4$ portions, and aggregate the fitness achieved in these positions (i.e. the total number of tiles visited in this region). This leads to a four-dimensional objective vector, where each objective ranges from 0 to 16. In this work, we study objective counts of $1, 4, 16, 64$. For our MAP-Elites implementation, we use the measure function defined as the end position of the knight before a rule break, and we use the aggregated score as the objective value.

### 4.2 MAZE

The Maze domain (Lehman & Stanley, 2011a) has often been cited as the motivating problem for novelty search based algorithms. In this maze, there is an agent represented by a neural network policy $\pi_\theta(s)$. Appendix A includes more details regarding this policy and other environmental configurations. The goal for this agent is to reach the top left of the maze, where myopically minimizing the distance to the goal would result in the agent falling into one of many local minima on the maze.

Other versions of this domain (Mouret & Doncieux, 2012; Grillotti & Cully, 2023) are presented with a slightly different purpose: to measure the "illumination" strength of these algorithms. This is a measure of how well the given algorithm can effectively spread across a given behavioral feature space. To do this, we must define the behavioral measures to be the end position of the robot. The objective then becomes some measure of the energy taken to reach this end position. A well performing illumination algorithm should find a large number of low energy consuming policies that result in reaching highly diverse parts of the maze. We decided to use this maze due to its historical precedence, as well as the fact that, with slight modifications to the reward function, it can be used as both a deceptive domain or an illumination domain.

**Objective for Deceptive Maze**    The objective in this task is to minimize the final distance to the goal. Another way to look at it is the agent must *maximize* the distance *moved towards the goal* at each time step. $r_t = d_t - d_{t-1}$ where $d_t$ is the distance from the agent to the goal at timestep $t$.

**Objective for Illumination Maze**    The objective in this task is based on that used by Grillotti & Cully (2023). At each timestep, the agent receives a reward based on the energy consumption during that time-step. The agents actions $\mathbf{a}_t$ can be characterized as:

$$\mathbf{a}_t = \text{clip}\left(\begin{bmatrix} \text{Velocity command wheel 1} \\ \text{Velocity command wheel 2} \end{bmatrix}, -1, 1\right) \times 0.025$$

and therefore, can assign the reward at each time step to be $r_t = -||\mathbf{a}_t||_2^2$. In order to get the total objective value or fitness $f$, you can simply aggregate this reward over every time step $f = \sum_{t=1}^{T} r_t$.

**Subaggregation** We follow the same subaggregation scheme for both types of maze domain. Since we get a single reward value per time-step, as well as a position of the agent at that time step, we follow a similar aggregation to that used in the knight's domain. We perform a quadtree subaggregation of the rewards received when the agent is in certain parts of the state space. For example, when we have four objectives, we aggregate the reward received by the agent in each quadrant of the board, and end up with a four dimensional objective function. $f_i = \sum_{t=1}^{T} \mathbb{1} [\text{Agent in quadrant } i \text{ at time } t] \times r_t$ where $\mathbb{1}$ is the indicator function, that evaluates to 1 if the agent is in quadrant $i$ at time $t$, or 0 otherwise. Since the agent can only be in one quadrant at a time, it becomes clear that $f = \sum_{i=0}^{n} r_i$ where $n$ is the number of objectives previously decided on, which satisfies our definition of a space based subaggregation. For both mazes, the measure function used for MAP-Elites is the end position of the robot, and the objective is the aggregated reward over the course of the trajectory.

## 5 EXPERIMENTS

We perform a series of experiments to empirically validate the performance of our implicit diversity preservation algorithm. For each of the lexicase selection based systems, we instantiate a single population of solutions, and repeatedly select from them based on the objective subaggregation that is in use. Then, we perform mutation on the population members. These populations have a fixed size $p$, and at each generation, we select (with replacement) $p$ individuals to be parents for the next generation. Note that this is different to the usual QD frameworks, where an archive of solutions is maintained over the course of the entire algorithm run. For the MAP-Elites based systems, we simply initialize an empty archive, and insert a series of solutions into the archive. Then, we select a batch of the elites from the archive, mutate them, and re-insert them if the cell that corresponds to their measure ($x$ and $y$ position for both domains) is empty or contains a worse solution. In these experiments, we set the batch size to be equal to the population size ($p$) used in lexicase selection.

We hold a number of things constant for this investigation. We hold the total number of individual evaluations, solution representation and mutation scheme constant across both systems. Since lexicase selection operates on a population, we maintain an archive similar to that used in the MAP-Elites loop to report all of our statistics from. Every generation, we insert the current population of solutions into the archive in the same way as detailed by MAP-Elites, and use this for comparison.

From these successive archives, we report three different kinds of statistics. Best Score refers to the score that the best individual in the population/archive at that step has achieved. QD-Score is simply defined as the sum of the scores for each individual in the archive, corrected to account for negative values. Finally, coverage is simply the percentage of the cells in the archive that are full. For each of these metrics, we report the mean over 10 replicate runs, with standard deviation shaded at every step of our search procedure. Below, we outline the results of these experiments.

### 5.1 KNIGHTS TOUR

For this domain, we report the Best Score and the QD-Score over time for both algorithms. We did not include coverage results for this domain as all algorithms had full coverage of the defined measure archive for every iteration. We find that for the Knight's tour domain, lexicase based approaches reach a solution that has a longer tour than MAP-Elites. This provides evidence that the MOO based approach is better at finding high performing solutions in the domain of choice. In terms of QD-Score, we also find that the lexicase selection based approaches outperform the MAP-Elites algorithm. The results for this can be found in Figure 1a. This was a surprising result, as MAP-Elites operates directly to maximize this value, whilst lexicase selection is agnostic to the measures we use to measure diversity. This hints that lexicase selection is able to maintain diversity with respect to many human-chosen diversity metrics, demonstrating its' potential applicability across a wide variety of domains.

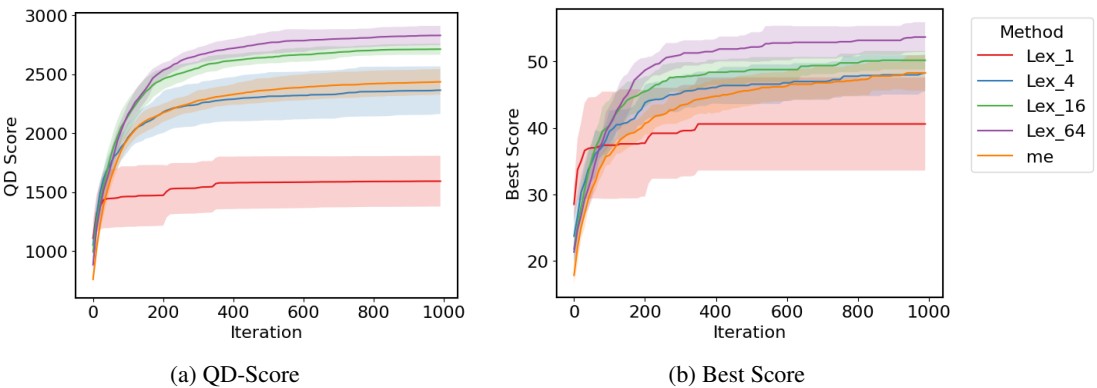

(a) QD-Score

(b) Best Score

Figure 1: Knight's Tour results. We compare implicit diversity preservation approaches using lexicase selection and MAP-Elites across iterations. Lexicase selection (lex_$n$ where $n$ is the number of objectives) shows an increasing performance given more objectives and outperforms MAP-Elites (me) with 16 and 64 objectives.

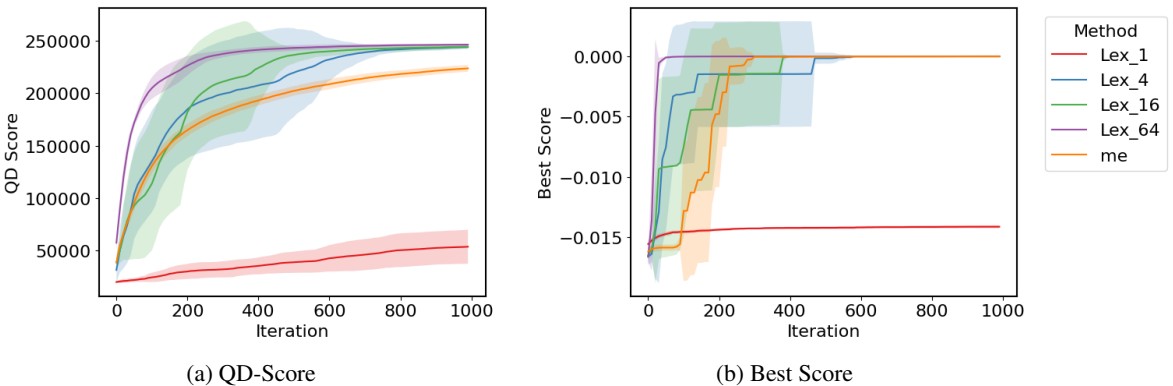

(a) QD-Score

(b) Best Score

Figure 2: Deceptive Maze results. Implicit diversity preservation techniques have a higher QD-Score than the archives for MAP-Elites. They also find the best solution faster than MAP-Elites.

## 5.2 MAZE

**Deceptive Maze**    The results for our deceptive maze can be found in Figure 2. From these figures, it is clear that the objective-based techniques are outperforming MAP-Elites. With more objectives, we see better performance on this domain. This is likely due to the added diversity preservation that arises from a large number of objectives, which is one of the benefits of lexicase selection. We also see that, with all techniques but Lex-1 (single objective optimization), a policy is found that overcomes the deception of the maze and actually achieves the globally optimal score of $0$. Surprisingly too do we see that the implicit diversity preservation techniques have a higher QD-Score than the archives for MAP-Elites, despite the fact that MAP-Elites is operating directly to optimize this metric.

**Illumination Maze**    The results for the illumination maze can be found in Figure 3a. We can see from these results a very similar conclusion to the prior two domains. Note that the objective is not designed to be deceptive and all algorithms very quickly reach the optimal score of $0$ (can be seen in Appendix C.2). However, when it comes to the QD Score, it appears that lexicase selection with 64 objectives actually outperforms MAP-Elites. This result was especially surprising for us as this domain is not inherently deceptive, and so using an objective-based algorithm would not necessarily make sense in this context. However, we still find that the space-based subaggregation prioritizes exploration in the search space implicitly. We study whether this improvement holds across subaggregation strategies in our ablation study in Section 6.

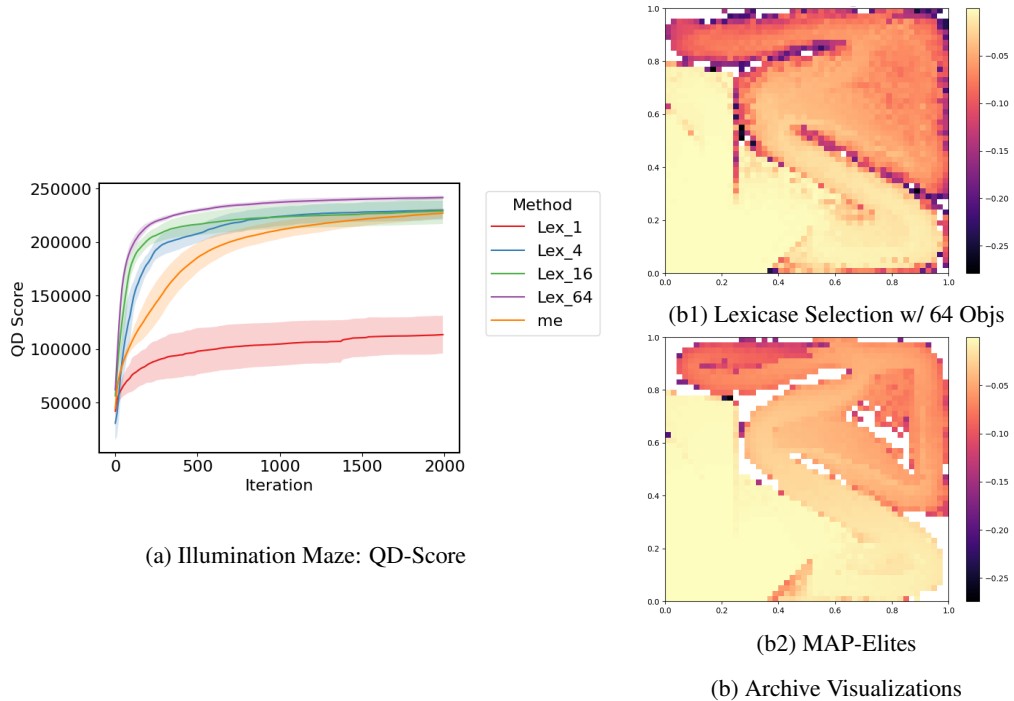

(a) Illumination Maze: QD-Score

(b1) Lexicase Selection w/ 64 Objs

(b2) MAP-Elites

(b) Archive Visualizations

Figure 3: Illumination Maze Archive results. (a) The QD-Score over time for a variety of methods. (b1) and (b2) show visualizations of the final archives for lexicase selection and MAP-Elites. A blank cell means that no individual was found with that given measure.

We also include a visualization of the archive to highlight the actual illumination abilities of each algorithm. Each position in the archive represents an individual that reaches that spot in the maze, and the fitness is determined by the energy used by the individual. Figures 3b1 and 3b2 shows two of the final archives from a randomly chosen replicate run. We can see that lexicase selection results in a better illumination of the measure space as more of the archive is filled with high performing solutions. The coverage results for both mazes are included in Appendix C.1. The results for coverage align well with the QD-Score metrics, as these two results are nearly directly correlated.

# 6 ABLATION STUDY

In order to verify the robustness of our proposed methodology, we ablate the subaggregation we used on both the deceptive and illumination maze, and we compare the results. For this experiment, we replace the subaggregation used on the maze domain from a space-based one to a time-based one. This means, instead of the multiple objectives referring to the objective value collected at different locations on the maze, the multiple objectives refer to the time step that these values were collected in. For example, with 4 objectives, the time based aggregation would simply sum the rewards achieved in the first, second, third and fourth quarters of time-steps of the episode, treating each as a single objective. This could be applied in any time-based RL domain.

The results of this ablation can be found in Figure 4. We see that the time-based and space-based subaggregation are competitive when the domain is deceptive. This is because in the deceptive domain, the ultimate goal is to maximize an objective. Maintaining diversity generates stepping stone solutions to the problem. As long as the subaggregation scheme used is reasonable, lexicase selection would result in added diversity preservation. We see that our algorithm maintains diversity the lines up with human intuitions of diversity as evidenced by the increased QD-score on the deceptive domain. However, when the main goal in the space is illumination of a certain set of qualitative measures, one should try to design a subaggregation scheme with these measures in mind. Without it, using an uncorrelated subaggregation scheme might result in low exploration with respect to these measures, as one would expect.

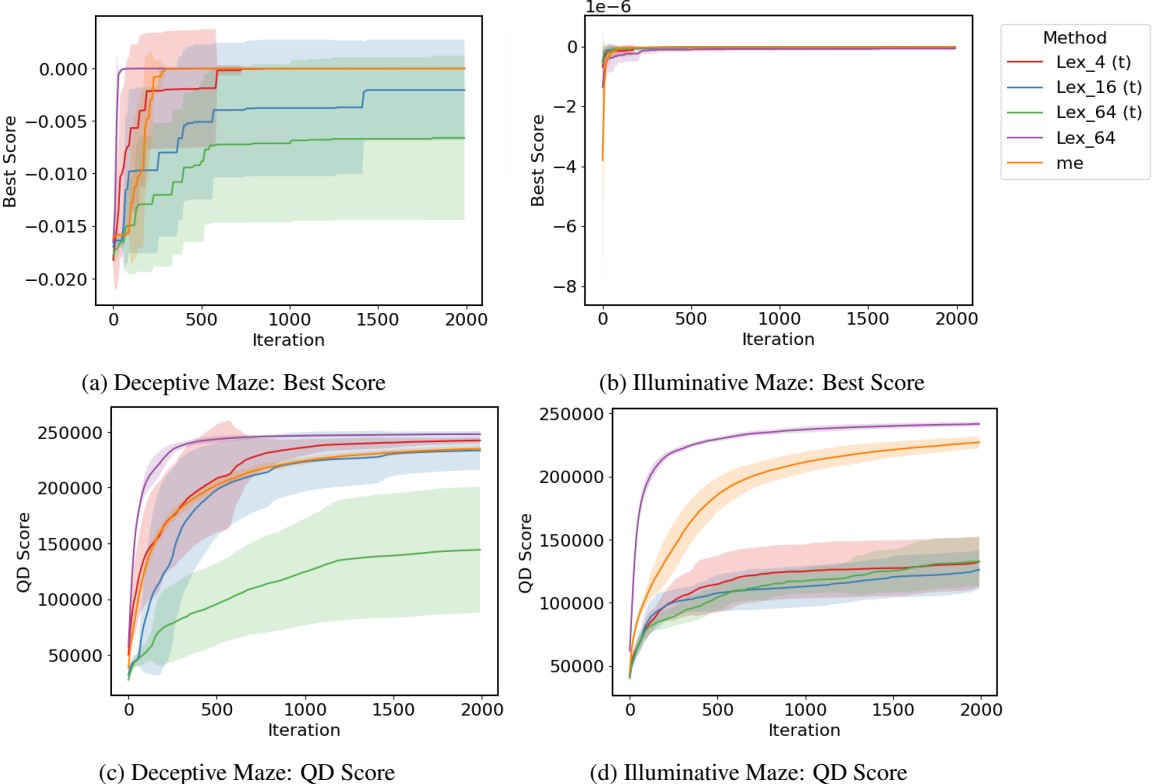

(a) Deceptive Maze: Best Score

(b) Illuminative Maze: Best Score

(c) Deceptive Maze: QD Score

(d) Illuminative Maze: QD Score

Figure 4: Results of the ablation study. We compare time-based subaggregation with 4, 16 and 64 objectives to the space based subaggregation results we show above with 64 objectives and to MAP-Elites (me). When comparing problem solving performance (or the best score achieved throughout the search process), it is clear that lexicase is robust to the choice of subaggregation when using a deceptive domain.

# 7 CONCLUSION

In this work, we presented a technique to solve potentially deceptive problems without explicitly measuring or maintaining diversity. Given a domain, we sub-aggregate the objective as opposed to fully aggregating it, and solve this subaggregation of objectives with many objective optimization. We use lexicase selection perform this optimization, which has demonstrated implicit diversity preservation properties whilst also being able to operate on large numbers of objectives.

We show that better search performance can be achieved using the objective-based approach than with diversity-based methods such as MAP-Elites. This suggests that when domains are deceptive, using implicit diversity preservation techniques can result in better stepping stones for optimization. Furthermore, we find that our objective-based approaches outperform the diversity-based approaches on evaluation metrics that are diversity-focused. Despite the fact that MAP-Elites is designed to optimize for QD-Score, our technique often performs better without directly optimizing these quantities. This demonstrates the wide applicability of our strategy to solve other deceptive domains.

Our ablation study shows that our technique can be robust to changes in the subaggregation scheme for deceptive domains. However, we find that when the domains are non-deceptive, the exploration ability of our algorithm is subject to the subaggregation scheme used, and likely would require more careful choices regarding how to aggregate states (based on their location spatially or temporally).

Lehman & Stanley (2011a) end their seminal novelty search paper with the line "To achieve your highest goals, you must be willing to abandon them." This work hints at an alternative view: "To achieve your highest goals, you must be willing to decompose them and work towards each component, in a potentially random order."

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

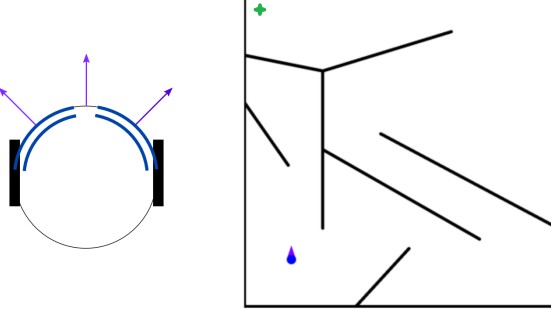

Figure 5: Kheperax domain adapted from Grillotti & Cully (2023)

---

**Algorithm 2:** Batch MAP-Elites, adapted from Mouret & Clune (2015)

**Data:**
- `env` - the environment to solve
- `measures()` - function that get descriptors

**Result:**
- An archive of individuals

$\mathcal{X} \leftarrow$ randomly initialize empty archive with $p$ random solutions
**for** `iter` $1 \rightarrow N$ **do**
    `candidates` $\leftarrow$ randomly select $p$ solutions from $\mathcal{X}$
    `scores` $\leftarrow$ sum(Evaluate(`candidates`, `env`)
    **for** $i : i \rightarrow p$ **do**
        $x_i \leftarrow$ randomly select $p$ candidates from archive
        $x'_i \leftarrow$ randomly mutate $x_i$
        $m'_i \leftarrow$ `measures`$(x'_i)$
        $f'_i \leftarrow$ use sum of scores $x'_i$
        $\mathcal{P}(m'_i) \leftarrow$ performance of elite in archive position for $m'_i$
        **if** $\mathcal{P}(m'_i) =$ or $\mathcal{P}(m'_i) < f'_i$ **then**
            $\mathcal{P}(m'_i) \leftarrow f'_i$
            $\mathcal{X}(m'_i) \leftarrow x'_i$
        **end**
    **end**
    **return** *Archive* $\mathcal{X}$
**end**

---

## A    MAZE DOMAIN SPECIFICATION

Figure 5 shows the Maze domain used from Grillotti & Cully (2023). In this domain, each individual is a neural network policy $\pi_\theta$ that encodes the control strategy for the robot pictured in the figure (left). There are three limited-range lasers placed at $-\frac{\pi}{4}, 0$ and $\frac{\pi}{4}$. We use a single MLP with 8 neurons in the hidden layer as the policy. In total, the episode lasts 250 timesteps.

## B    MAP-ELITES ALGORITHM

An outline of the MAP-Elites (Mouret & Clune, 2015) algorithm can be found in algorithm 2. Given an archive, and a "measure" function that outputs the qualitative features for each individual, MAP-Elites continually inserts and selects individuals from this archive based on their fitness score and their measures. This results in exploration (as new cells that are reached can have a solution of any quality inserted), as well as exploitation (as if an individual improves upon the fitness of an another while achieving the same measures, it replaces the other in the archive).

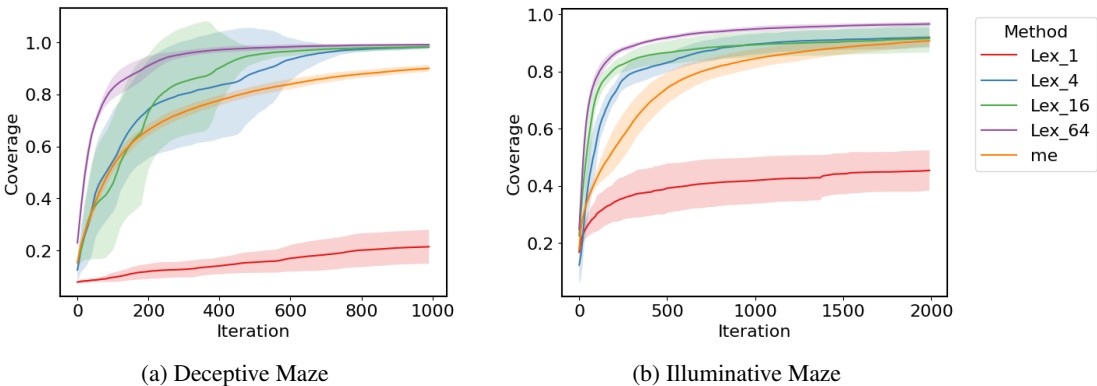

(a) Deceptive Maze                    (b) Illuminative Maze

Figure 6: Percentage coverage of the mazes by solutions discovered by each algorithm.

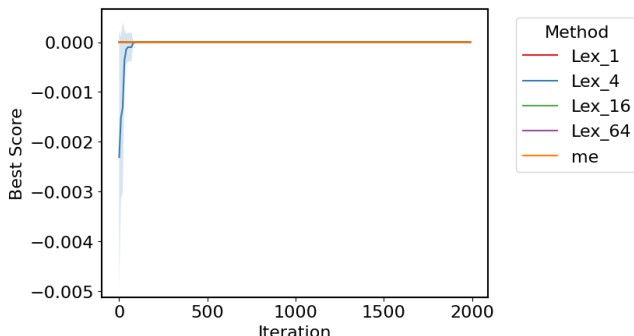

Figure 7: Illumination Maze: Best performance achieved by each algorithm

## C  FURTHER RESULTS FOR MAZES

### C.1  COVERAGE RESULTS FOR THE TWO MAZE DOMAINS

We include results (Figure 6) for the coverage of the archive for each of these algorithms. The coverage is the fraction of the archive that is full at every step of the search procedure. For the deceptive domain, these results are directly proportional to the QD-Score results as the objective value at a given portion of the archive is entirely determined by it's $(x, y)$ position. For the illumination maze, whilst not directly proportional, we expect the coverage to be highly correlated with the QD-Score, as discovering more solutions would lead to a higher QD-Score as well as coverage.

### C.2  BEST SCORE FOR ILLUMINATION MAZE

We include results (Figure 7) for the best performance of any individual produced using MAP-Elites versus those produced by Lexicase selection. It is clear that both of these algorithms rapidly discover solutions with the minimum energy expenditure of 0. This highlights the fact that this domain is not deceptive at all, as achieving a globally optimal solution is trivial (just staying still suffices to achieve this).

## D  ARM REPERTOIRE

In order to verify the performance of our algorithm in domains that are not deceptive, we include results from a new domain. The arm domain (Vassiliades & Mouret, 2018) is one specifically designed to test Quality Diversity algorithms' illumination ability. The goal represented by this domain is to find the angle parameters of a multiple link arm that leads to low variance between the angles,

whilst also reaching a given $(x, y)$ position with the end-effector. The implementation was based off of code by Tjanaka et al. (2021).

**Objective and Subaggregation** The objective in this domain is to have the lowest possible variance of the parameters $\theta$ of the arm. Therefore, we define the fitness of the arm to be the negated variance of the angle parameters. The subaggregation we use here is very simple: we take the variance of $n$ subsets of the angles in the arm. This is most similar to a time-based de-aggregation, although this domain is not time-based. For example, with an 8-link arm, and 4 objectives, we split the angle values into 4 lists of length 2, and find the total variance from each pair of angles to the mean of the entire arm.

## D.1 RESULTS

We study arms with 8, 16 and 32 links (figures 8, 9, 10, respectively). Across these three configurations, we see that, for this non-deceptive domain, our implicit diversity preservation technique outperforms MAP-elites with respect to the actual raw score. This makes sense, as when there is no diversity preservation needed in the domain, simply following an objective is the best way to reduce the variance between the links of the arm.

However, across all three of the parameterizations of the domain, MAP-Elites significantly outperforms out implicit diversity maintenance technique when it comes to QD-Sccre. This is likely due to the fact that the domain is not deceptive at all, and as such the diversity maintenance is not instrumental at all. In other words, there is no need to explore low-fitness portions of the search space, as those do not lead to global optima in this domain.

## D.2 ARCHIVE VISUALIZATIONS

In order to validate the conclusions, we visualize the resulting archive from using our implicit diversity maintenance techniques as well as MAP-Elites. These figures depict all the points in search space that are illuminated by each algorithm over the course of it running. We select these archives from the final iteration of a random run of each of the studied techniques. These archives are depicted in figures 11, 12, 13 for the 8-, 16- and 32-link arm, respectively.

From these figures, we can conclude that the implicit diversity maintenance technique illuminates the high-quality portions of the domain, but does not explore the regions with low fitness. However, MAP-Elites explores all regions regardless of their quality. This result further validates that, when domains are not deceptive, our implicit diversity maintenance technique prioritizes exploitation over exploration, which indeed results in higher quality solutions. However, the technique does not "illuminate" the space of solutions, as MAP-Elites does.

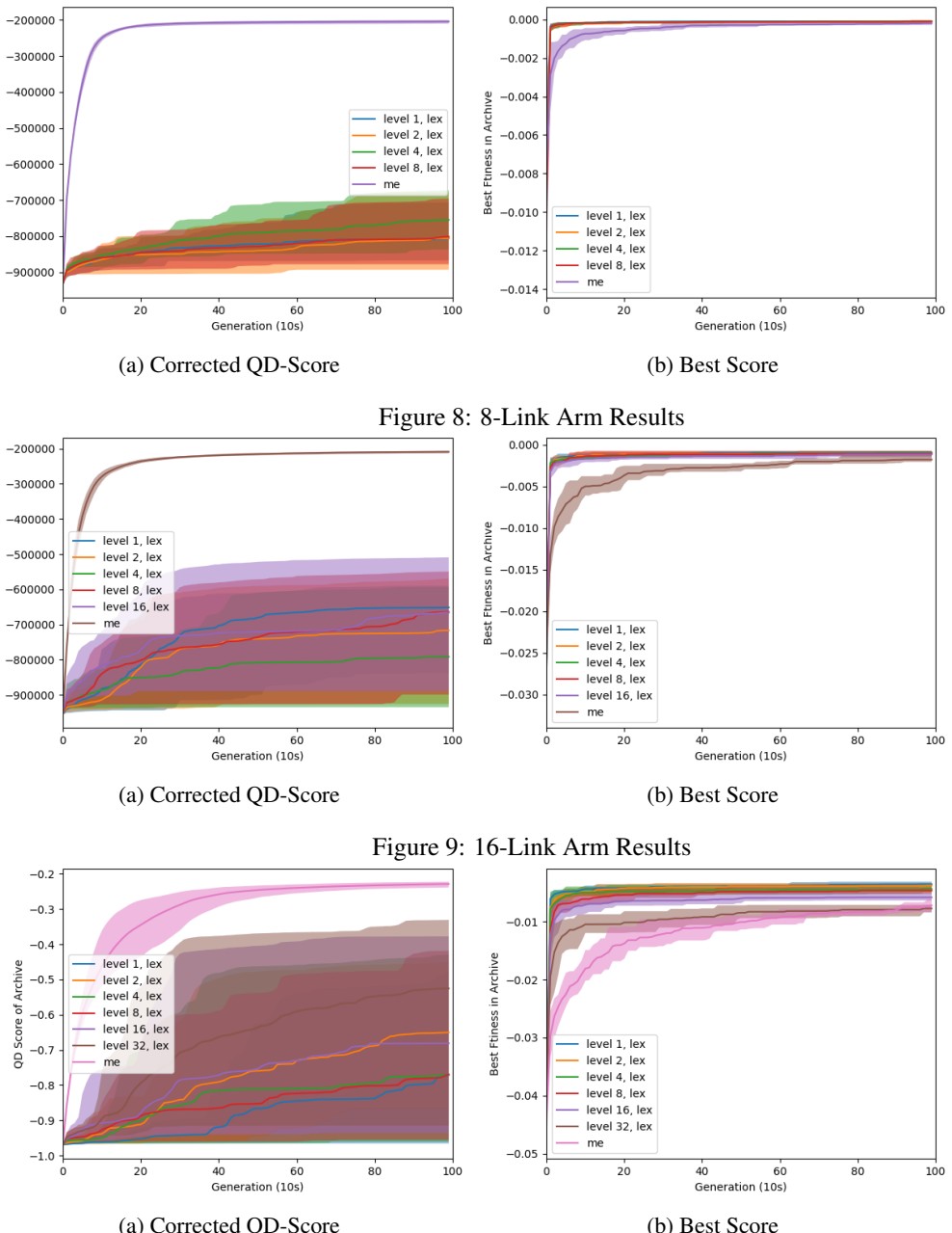

(a) Corrected QD-Score

(b) Best Score

Figure 8: 8-Link Arm Results

(a) Corrected QD-Score

(b) Best Score

Figure 9: 16-Link Arm Results

(a) Corrected QD-Score

(b) Best Score

Figure 10: 32-Link Arm Results

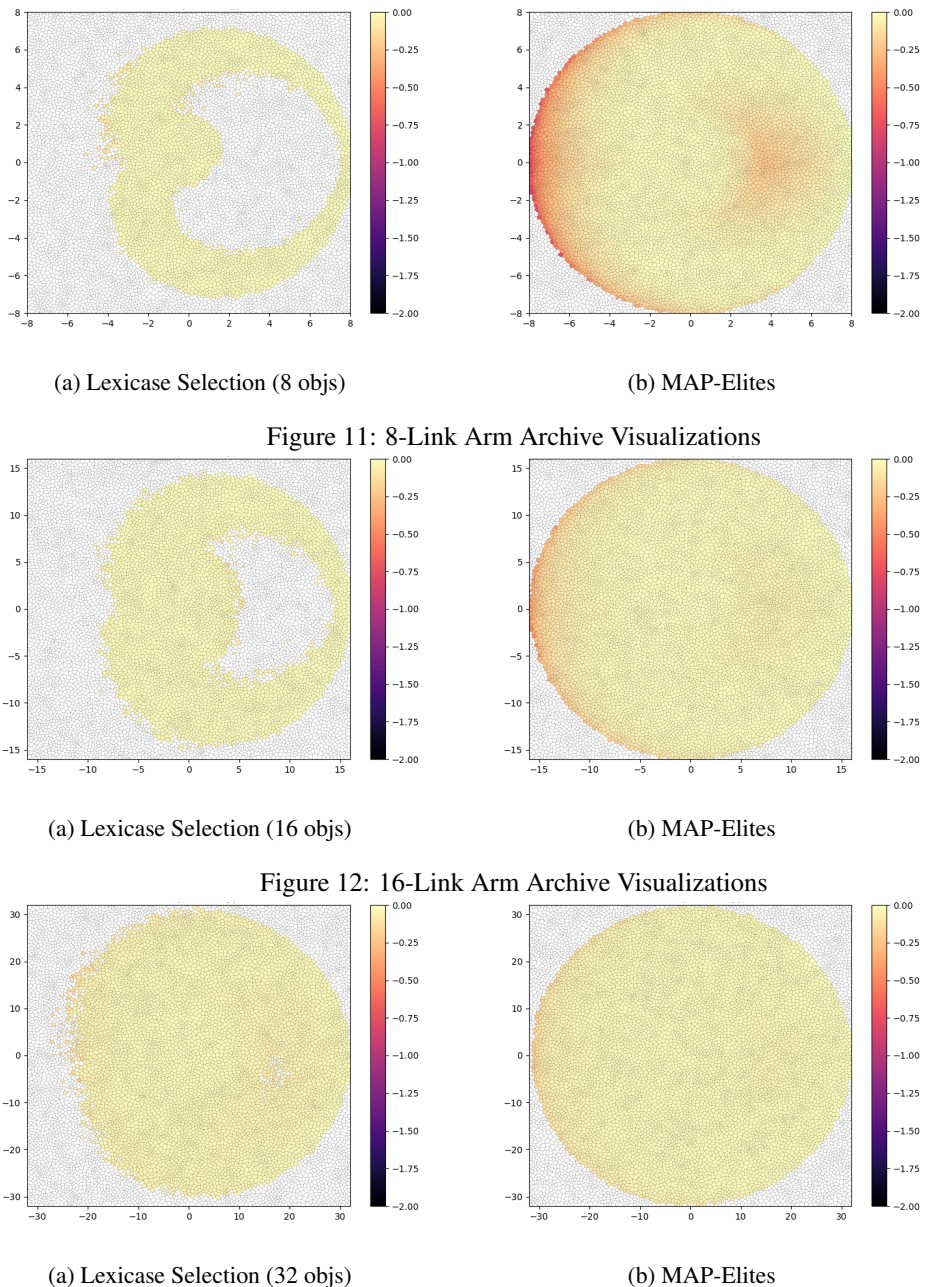

(a) Lexicase Selection (8 objs)        (b) MAP-Elites

Figure 11: 8-Link Arm Archive Visualizations

(a) Lexicase Selection (16 objs)        (b) MAP-Elites

Figure 12: 16-Link Arm Archive Visualizations

(a) Lexicase Selection (32 objs)        (b) MAP-Elites

Figure 13: 32-Link Arm Archive Visualizations

