# OpenReview forum: "Objectives Are All You Need: Solving Deceptive Problems Without Explicit Diversity Maintenance"
_ICLR.cc/2024/Conference — Submitted to ICLR 2024_

### Official Review · Reviewer_sSB3 · 2023-10-28

**Soundness:** 2 fair
**Presentation:** 2 fair
**Contribution:** 2 fair
**Rating:** 3
**Confidence:** 2

**Summary:**

Some search algorithms can achieve improved performance by promoting exploration through the encouragement of higher diversity within their solution pool. Traditionally, achieving this has necessitated the manual engineering of appropriate diversity measures and incorporating them explicitly in the algorithm. In this paper, the authors introduce a new approach: they transform the original single objective into multiple objectives for the search algorithms, which promotes better exploration as well. This strategy produces superior results in the two domains that present deceptive challenges for search algorithms when compared to the performance of MAP-Elites.

**Strengths:**

The comparison between the proposed algorithm and the MAP-Elites algorithm has been fair and the results have been analyzed carefully and thoroughly.

**Weaknesses:**

The range of problems where the proposed algorithm would be useful, as covered by the two example problems given in the paper, are too narrow. Both problems take place on a 2D plane where the space-based subaggregation method can be applied naturally. I would like to see the proposed method applied in broader domains.

**Questions:**

For example, can this approach be used in solving Travelling Salesman Problems?

---

> ### Author Response · Authors · 2023-11-21
> **Response to Reviewer sSB3**
>
> Dear reviewer sSB3,
>
> Thank you for taking the time to review our work. Your comments have helped us improve our paper as well as understand its strengths and weaknesses.
>
> We respond to your comments below. Please let us know if there are any lingering questions that might be getting in the way of an acceptance recommendation.
>
> > W1: The range of problems where the proposed algorithm would be useful, as covered by the two example problems given in the paper, are too narrow. Both problems take place on a 2D plane where the space-based subaggregation method can be applied naturally. I would like to see the proposed method applied in broader domains.
>
> Thank you for this suggestion. We agree that we must detail the conditions under which this algorithm can work well. We believe that our method can work in problems regardless of their dimensionality. We have studied 1D (time-based subaggregation) and 2D (space-based subaggregation), but we do not see why this method cannot easily be applied to higher dimensions by simply changing the way we sub-aggregate to be sensitive to the new dimensions. Furthermore, we believe there are many interesting problems that exist on a 2D plane that would be interesting to study in future work. For example, Go, Chess, and all the Atari benchmarks have only two dimensions in state space, yet would each come with their own unique challenges and deceptive-ness.
>
> Furthermore, we have added a new domain to the paper, which involves finding parameter values for a robotic arm. This is another example of a problem that exists on a 2D plane, however we think it presents new insights due to the structure of the search landscape.
>
> > Q1: For example, can this approach be used in solving Travelling Salesman Problems?
>
> This is a great connection to our work. The Knight’s tour domain can be thought of as a specific case of the Travelling Salesperson problem, or more specifically, a Hamiltonian path problem, as there is no requirement to return to the starting position. The Knight's tour problem trivially reduces to the Hamiltonian path problem, and is therefore also NP-Hard. In general, we think that, with limited modifications, our algorithm can trivially extend to solving the more general case of Hamiltonian path or other TSP-like problems. By defining the sub-aggregation with respect to the “cities”, we could easily maintain all the fruitful visitation orderings that we have found so far, which would ultimately help solve the problem, which is indeed very deceptive.
>
> Thank you once again for your unique insights and connections made to our work. We would be happy to answer any further comments or concerns you have!

---

> > ### Comment · Reviewer_sSB3 · 2023-11-21
> >
> > Thank you for your response. My primary concern remains that the methodology presented may be effective exclusively for the problems specifically outlined in the paper. I believe that demonstrating the method's applicability to a wider array of problems would substantially enhance the paper's strength and impact.
> > I will maintain my current review score due to these ongoing concerns.

---

### Official Review · Reviewer_5f3M · 2023-11-01

**Soundness:** 3 good
**Presentation:** 2 fair
**Contribution:** 2 fair
**Rating:** 3
**Confidence:** 4

**Summary:**

This paper presents an approach to avoid getting stuck in local optima in deceptive domains by optimizing a large set of automatically extracted objectives. While most approaches in evolutionary computation to avoid deception are based on explicitly rewarding diversity, like novelty search or quality diversity, the presented approach directly optimizes for a set of objectives. It is evaluated on two domains: knight’s tour and a deceptive 2D maze.

**Strengths:**

- Automatically extracting objectives from the environment is a promising approach to elivate some of the issues with many quality diversity algorithms
- Promising results in two deceptive domains

**Weaknesses:**

- Easy read for somebody in EA community but should be more motivated for why the broader ML community should care.
- MAP-Elites is not introduced when first mentioned in the introduction
- This description should be extended to make it very clear for the reader "The idea of lexicase selection is that instead of compiling performance metrics over the training dataset, we can leverage individual performance to sift through a population via a sequence of ran domly shuffled training cases.”
- More complex domains should be tested and the approach should be compared to RL-based approaches as baselines to increase its impact on the broader ML community

Minor comment:
- I would suggest not another “all you need” title...
- It should be noted in Figure 2 that n is the number of objectives in lex_n. Is lex_1 equivalent to not having any lexicase, just a standard selection?
- Add a description of lexicase to the abstract/introduction

**Questions:**

- Can this approach be extended to RL methods, which would significantly increase its impact?
- How would the reward aggregation work in higher-dimensional domains, i.e. learning directly from pixels? Would it be necessary to manually define the objectives in that case?

---

> ### Author Response · Authors · 2023-11-21
> **Response to Reviewer 5f3M**
>
> Dear Reviewer 5f3M,
>
> Thank you very much for your comments on our paper. We appreciate the positivity in your review as well as the advice you have given us to improve. We respond to each of your worries in turn.
>
> > W1: Easy read for somebody in EA community but should be more motivated for why the broader ML community should care.
>
> Thank you for pointing this out. We have added a more in depth description of our evolutionary algorithm used here to make this work more broadly understandable to people unfamiliar with the field. Furthermore, we have added an algorithm outlining our entire proposed system to help a reader follow along in more depth and to aid in reproducibility. We also include an algorithm to our appendix outlining the algorithm we are comparing to, MAP-Elites.
>
> > W2: MAP-Elites is not introduced when first mentioned in the introduction
>
> Thank you for the comment. We agree that we did not sufficiently explain MAP-Elites as a baseline. We briefly introduce the algorithm in the main text to help with readability and we also provide a detailed description in the appendix for a curious reader to learn more.
>
> > W3: This description should be extended to make it very clear for the reader "The idea of lexicase selection is that instead of compiling performance metrics over the training dataset, we can leverage individual performance to sift through a population via a sequence of randomly shuffled training cases.”
>
> Thank you for suggesting this. We have included an algorithm outlining our entire procedure (including the lexicase selection portion), and improved our in text explanation to better outline the lexicase selection procedure.
>
> > W4: More complex domains should be tested and the approach should be compared to RL-based approaches as baselines to increase its impact on the broader ML community
>
> We appreciate this comment. We agree and believe that this technique can be applied to more complex RL-based approaches to help benchmark it in a more rigorous manner. However, we believe that the conclusions from this paper are broadly applicable due to the variety of domains (discrete optimization for knight touring and RL for the maze) studied. To help further demonstrate this, we add a fourth domain, the robotic arm, to the appendix. We hope that this work can motivate future work to try our implicit diversity maintenance in more advanced RL domains such as Atari.
>
> > MW1: I would suggest not another “all you need” title...
>
> Thanks to your suggestion, we have opted for a new title, that expressed the conclusions from this paper well: "Deceptive Problems Can Be Solved Without Explicit Diversity Maintenance"
>
> > MW2:  It should be noted in Figure 2 that n is the number of objectives in lex_n. Is lex_1 equivalent to not having any lexicase, just a standard selection?
>
> Yes, lex_1 is equivalent to not performing lexicase selection. In fact, it is equivalent to best-of-n hill climbing, where we simply select the best individual in the entire population based on the aggregated fitness value. We have added an explanation of the naming convention to the figure caption.
>
> > MW3: Add a description of lexicase to the abstract/introduction
>
> We agree with this idea and have added a short description of lexicase selection to the introduction. We hope that this, in combination with our included algorithm, will sufficiently outline the lexicase selection algorithm
>
> Thank you very much for including these constructive comments. They have been very helpful in improving the quality of our work!

---

> ### Author Response · Authors · 2023-11-21
> **Response to the Questions of Reviewer 5f3M**
>
> Please find below some of our answers to your questions:
>
> > Q1: Can this approach be extended to RL methods, which would significantly increase its impact?
>
> Thanks for the great question. In fact, the maze domains that we use requires performing search over parameters for a neural network policy, which is an evolutionary approach to performing RL. We have added specific details including neural network policy parameters to the appendix. For this reason, we strongly believe that this approach is compatible and broadly applicable to higher dimensional RL methods. As such, we believe that scaling up our technique to higher dimensional RL domains is trivial once a sub-aggregation is determined.
>
> > Q2: How would the reward aggregation work in higher-dimensional domains, i.e. learning directly from pixels? Would it be necessary to manually define the objectives in that case?
>
> This is a great question. There has been some prior work discussing where to get measures from when there is no human in the loop (or no domain knowledge) to create them. There has been recent work in the QD literature that surrounds learning these metrics of qualitative difference directly from human preferences as is done in the following paper:
>
> Ding, Li, et al. "Quality Diversity through Human Feedback." arXiv preprint arXiv:2310.12103 (2023).
>
> And very famous work that involves learning these measures directly from the environment with an autoencoder:
>
> Cully, A. (2019, July). Autonomous skill discovery with quality-diversity and unsupervised descriptors. In Proceedings of the Genetic and Evolutionary Computation Conference (pp. 81-89).
>
> Finally, there has been work that attempts to extend the autoencoder system to objective based optimization algorithms:
>
> Boldi, R., & Spector, L. (2023). Can the Problem-Solving Benefits of Quality Diversity Be Obtained Without Explicit Diversity Maintenance?. arXiv preprint arXiv:2305.07767.
>
> These recent papers propose unsupervised and semi-supervised methods to learn the diversity metrics instead of manually defining them. Our method can be easily combined with these techniques to aggregate reward with respect to many different metrics. There are also many techniques for self-supervised feature extraction for RL environments, such as those learned by predicting forward or backward dynamics models, or even reward functions.
>
> We hope that these responses answer your questions. Feel free to ask if any clarification is needed on any of these points.
>
> Thank you for the time you've taken to review our paper. If you think that our responses to your questions (this comment) and concerns (earlier comment) were satisfactory, we ask that you consider raising your score.

---

> > ### Comment · Reviewer_5f3M · 2023-11-22
> >
> > Thank you for the clarifications. I still do believe that evaluating on more complex domains would be important to show the generality of the method, so I will keep my original score for now. But I'm very much looking forward to seeing the approach developed further in the future.

---

### Official Review · Reviewer_oe8p · 2023-11-03

**Soundness:** 3 good
**Presentation:** 3 good
**Contribution:** 3 good
**Rating:** 6
**Confidence:** 4

**Summary:**

This paper introduces an approach based on objective sub-aggregation designed to address potentially misleading domains by optimizing a set of specified objectives via lexicase selection, thereby converting the fundamental search problem into a Multi-Objective Optimization (MOO) challenge. The authors argue that by utilizing lexicase selection to optimize the sub-aggregated objectives, maintaining explicit diversity measures becomes unnecessary, as implicit diversity is ensured. This objective-driven method surpasses the performance of the cutting-edge quality diversity algorithm, MAP-Elites, specifically on misleading domains, achieving competitive results even though it does not explicitly prioritize diversity. Additionally, an ablation study confirms the reliability of the sub-aggregation technique by demonstrating that various sub-aggregation strategies produce comparable performance, showcasing the algorithm's adaptability.

**Strengths:**

1. The research explores a promising direction, illustrating the effectiveness of implicit diversity preservation in Multi-Objective Optimization (MOO) problems using lexicase selection. This approach is widely adopted for MOO tasks, facilitating diverse solutions without direct optimization for diversity.

2. The conducted experiments and ablation study offer a thorough examination. The selection of Knight’s Tour and Maze (both Deceptive and Illumination) domains is thoughtfully explained, including the specific objectives, sub-aggregation, objective counts, and other pertinent details.

3. The paper is well-structured with a coherent flow, especially evident in the well-crafted "Introduction" and "Related Work" sections that provide comprehensive and specific insights.

4. Comparative analysis based on Best Score, QD-Score, and coverage presents valuable insights into the proposed approach's effectiveness and implicit diversity preservation.

**Weaknesses:**

1. In Section 4.2 under “Objective for Deceptive Maze”, the first two lines of the paragraph seem contradictory.

2. The introductory paragraph of Section 3 mentions the limitations of explicit diversity maintenance and proposes
implicit diversity maintenance as a viable alternative. Instead of just a mention, a thorough explanation with
suitable illustrations would be better. For instance, consider the following line - “More importantly, in more complex or deceptive search spaces where the relationship between phenotypic traits and fitness is not straightforward, these explicit measures can
inadvertently steer the search away from optimal or even satisfactory solutions.” The author(s) could have
taken a particular deceptive domain, pictorially representing how the search deviates from the optimal and
how implicit diversity solves this issue.

3. Could deceptive domains from OpenAI Gym/Atari be included? Montezuma’s Revenge is a classic example!
The authors claim that objectives are all we need for solving any deceptive domain. A performance comparison
of the proposed approach to other SOTA algorithms in these domains would make the claim stronger.

4. The author(s) also claim that this approach enhances exploration in deceptive domains. However, it is also
important to evaluate the performance of RL agents (trained by the proposed algorithm) in non-deceptive
MuJoCo environments. This would validate the algorithm’s sensitivity to the choice of the sub-aggregation
schemes.

5. An algorithm is missing! The entire approach should have been formalized as an algorithm and written clearly
in the main paper.

**Questions:**

I would like the author(s) to address the points mentioned in the “Weaknesses” section.

---

> ### Author Response · Authors · 2023-11-21
> **Response to Reviewer oe8p**
>
> Dear Reviewer oe8p,
>
> Thank you very much for taking the time to craft a detailed and insightful review. We really appreciate your advice and ideas to help improve our paper. We would first like to sincerely thank you for the positive and constructive comments you have given us. We hope that our responses help resolve the concerns presented as well as answer some of the lingering questions.
>
> > W1:  In Section 4.2 under “Objective for Deceptive Maze”, the first two lines of the paragraph seem contradictory.
>
> Thank you for this comment. We have adjusted the text to remove the apparent contradiction.
>
> > W2: The introductory paragraph of Section 3 mentions the limitations of explicit diversity maintenance and proposes implicit diversity maintenance as a viable alternative. Instead of just a mention, a thorough explanation with suitable illustrations would be better. For instance, consider the following line - “More importantly, in more complex or deceptive search spaces where the relationship between phenotypic traits and fitness is not straightforward, these explicit measures can inadvertently steer the search away from optimal or even satisfactory solutions.” The author(s) could have taken a particular deceptive domain, pictorially representing how the search deviates from the optimal and how implicit diversity solves this issue.
>
> Thank you for the great suggestion. We should have included a motivating example to help a reader align their intuition with ours. We have added the following example to the paper:
> “Take a deceptive maze that has no exterior walls. Novelty search and MAP-Elites might devote too much time exploring the space outside the maze, whereas the implicit diversity preservation would only preserve this diversity if it is instrumental in finding the goal (I.e. probably inside the maze)”
>
> We also verify this point experimentally through the inclusion of a non-deceptive domain to our appendix, where optimizing for diversity is simply a distraction away from solving the (admittedly very trivial) objective. In this domain, our algorithm reaches a higher objective value than the method that explicitly maintains diversity.
>
> > W3 : Could deceptive domains from OpenAI Gym/Atari be included? Montezuma’s Revenge is a classic example! The authors claim that objectives are all we need for solving any deceptive domain. A performance comparison of the proposed approach to other SOTA algorithms in these domains would make the claim stronger.
>
> This is a great question! Yes, we believe that this paper can easily be extended to larger domains with more complicated search spaces. However, as an initial proof of concept, we believe that the relatively simple domains are good for the diagnostic experimentation that we performed, paving the way for future work to extend our claims to larger domains.
>
> > W4: The author(s) also claim that this approach enhances exploration in deceptive domains. However, it is also important to evaluate the performance of RL agents (trained by the proposed algorithm) in non-deceptive MuJoCo environments. This would validate the algorithm’s sensitivity to the choice of the sub-aggregation schemes.
>
> Thank you for the great idea. We have decided to include an entirely non-deceptive domain in our work to help demonstrate the sensitivity of our algorithm to non-deception. We would love to extend this to MuJoCo or other advanced RL domains in future work. Although the illumination maze is not inherently deceptive, we include another commonly used QD domain, known as the Arm Repertoire, that is entirely non-deceptive. In order to maximize the objective, the optimization algorithm should minimize the variance between angles of a 2D robotic arm. Our results show that when the domains are non-deceptive, our approach does not outperform MAP-Elites on the diversity focused metrics, but does outperform it on the quality-based metrics. This makes sense as the domains that are not deceptive are easy to maximize by simply following an objective, whereas other QD algorithms might spend too many resources exploring low quality regions of the search space (boosting the QD-Score). This further illustrates why our method works better than ME on deceptive tasks: it focuses more towards quality and uses diversity to drive search if/when it is needed, while ME equally balances diversity and quality regardless of their instrumentality in solving the problem at hand.
>
> > W5: An algorithm is missing! The entire approach should have been formalized as an algorithm and written clearly in the main paper.
>
> Thank you for pointing this out. We have added an algorithm detailing our approach in Section 3. We hope that this makes it easier to follow our proposal!
>
> In general, we would like to thank you once again for the help you have given us to improve our paper. If you think that our revisions and comments have addressed your questions and concerns, we kindly ask that you consider raising your rating.

---

### Meta-Review · Area_Chair_XfgV · 2023-12-05

**Metareview:**

This paper studies an objective maximizing baseline and compares it with QD algorithms in the domains where QD algorithms are usually evaluated on. Based on their results, the authors claim that maximizing diversity is not required in these domains. However, the reviewers pointed out that the results are specific to the domains studied, and it is not clear how the results will generalize to more complex domains. I hope that the authors will improve these aspects in the next version of their paper, but in its current form, its not ready for publication.

**Justification For Why Not Higher Score:**

Limited results in specific domain.

**Justification For Why Not Lower Score:**

N/a

---

### Decision · Program_Chairs · 2024-01-16

Reject